# A Unique P450 Peroxygenase System Facilitated by a Dual-Functional Small Molecule: Concept, Application, and Perspective

**DOI:** 10.3390/antiox11030529

**Published:** 2022-03-10

**Authors:** Siyu Di, Shengxian Fan, Fengjie Jiang, Zhiqi Cong

**Affiliations:** 1CAS Key Laboratory of Biofuels, and Shandong Provincial Key Laboratory of Synthetic Biology, Qingdao Institute of Bioenergy and Bioprocess Technology, Chinese Academy of Sciences, Qingdao 266101, China; disy@qibebt.ac.cn (S.D.); fansx@qibebt.ac.cn (S.F.); jiangfj@qibebt.ac.cn (F.J.); 2University of Chinese Academy of Sciences, Beijing 100049, China

**Keywords:** cytochrome P450 monooxygenase, peroxygenase, peroxidase, protein engineering, oxidation, hydroxylation, epoxidation, sulfoxidation, dual-functional small molecule

## Abstract

Cytochrome P450 monooxygenases (P450s) are promising versatile oxidative biocatalysts. However, the practical use of P450s in vitro is limited by their dependence on the co-enzyme NAD(P)H and the complex electron transport system. Using H_2_O_2_ simplifies the catalytic cycle of P450s; however, most P450s are inactive in the presence of H_2_O_2_. By mimicking the molecular structure and catalytic mechanism of natural peroxygenases and peroxidases, an artificial P450 peroxygenase system has been designed with the assistance of a dual-functional small molecule (DFSM). DFSMs, such as *N*-(ω-imidazolyl fatty acyl)-l-amino acids, use an acyl amino acid as an anchoring group to bind the enzyme, and the imidazolyl group at the other end functions as a general acid-base catalyst in the activation of H_2_O_2_. In combination with protein engineering, the DFSM-facilitated P450 peroxygenase system has been used in various oxidation reactions of non-native substrates, such as alkene epoxidation, thioanisole sulfoxidation, and alkanes and aromatic hydroxylation, which showed unique activities and selectivity. Moreover, the DFSM-facilitated P450 peroxygenase system can switch to the peroxidase mode by mechanism-guided protein engineering. In this short review, the design, mechanism, evolution, application, and perspective of these novel non-natural P450 peroxygenases for the oxidation of non-native substrates are discussed.

## 1. Introduction

Cytochrome P450s (CYPs or P450s), a broad class of heme-containing enzymes, play important roles in drug metabolism, detoxification of xenobiotics, and steroid biosynthesis [1]. These enzymes are ubiquitous in nature, being found in animals, plants, bacteria, fungi, and other organisms [2]. P450s have potential use in the catalytic monooxygenation of various organic substrates, including aliphatic and aromatic compounds, alkenes, and compounds containing heteroatoms such as nitrogen and sulfur [3,4,5,6,7,8]. In particular, P450s can regio- and stereoselectively oxidize inert C–H bonds, thus acting as an attractive enzyme class in the development of practical biocatalysts for organic synthesis [9,10,11].

A variety of approaches have been developed to solve the intrinsic drawbacks of P450s, e.g., poor enzyme stability, low turnover rates, narrow substrate scope, and the need for expensive cofactors (NAD(P)H). Protein engineering, including rational design and directed evolution, represents a first choice for solving most of these issues [12,13,14]. Moreover, effective strategies have emerged to overcome some specific problems during P450-driven catalysis [15,16,17,18,19,20,21,22,23,24,25,26,27,28,29,30,31,32,33]. For example, researchers have constructed a substrate engineering approach to improve the acceptance and/or the stereo-/regioselectivity of non-native substrates of P450s by introducing protecting/anchoring/directing groups to the substrate [15,16,17,18,19,20,21,22,23,24]. Watanabe and co-workers used a dummy co-substrate (so-called decoy molecule) to modulate substrate promiscuity of P450s, enabling wild-type P450s to hydroxylate various small molecules that are not accepted in the absence of the decoy molecule (such as gas alkanes and benzene) [25,26,27]. Biological and chemical regeneration of NAD(P)H has been used widely to support catalysis by P450s [28,29]. In addition, the surrogate oxidants (e.g., hydrogen peroxide, *tert*-butyl hydroperoxide, and iodosylbenzene) are also used to drive P450 instead of molecular oxygen and reduced NAD(P)H [30,31,32,33]. Despite being useful supplements to protein engineering, these strategies often play a role in combination with protein engineering. There have been many reviews discussing the abovementioned topics [34,35,36,37,38,39,40,41]. Herein, we focus on a unique H_2_O_2_-dependent P450 peroxygenase system facilitated by a dual functional small molecule (DFSM). The design, construction, mechanism, and catalytic application of the DFSM-facilitated P450 peroxygenases are reviewed, and current issues and future perspectives are also discussed.

## 2. Proof-of-Concept of the DFSM-Facilitated P450 Peroxygenase

The complex catalytic cycle of P450s needs reduced co-enzyme NAD(P)H and a redox partner to support the activation of molecular oxygen. Thus, it had been suggested that surrogate peroxide species can be used to drive P450 catalysis through its shunt pathway (Figure 1), with low-cost H_2_O_2_ being one of the best choices. However, only a few native P450 peroxygenases (e.g., CYP 152 family) can use the unique substrate-assisted mechanism to activate H_2_O_2_ successfully [42,43,44,45,46,47], with most P450s examined (e.g., rat liver microsomal P450, human P450s such as CYP1A2 and 3A4, thermophilic archaea CYP119, CYP175A1, and P450cam) generally showing low efficiency for the H_2_O_2_-dependent reaction (shunt pathway in Figure 1) [48,49,50,51,52,53]. Although the peroxygenase and peroxidase activity of P450s can be partially improved by directed evolution, the catalytic efficiency of the evolved P450 variants is still not comparable to natural NAD(P)H-dependent P450s [54,55,56,57]. This may be caused by the inherent structural characteristics of P450s. Indeed, those enzymes that make good use of H_2_O_2_ in nature have acid-base amino acid residue pairs that play the role of an acid–base catalyst in their active site (Figure 2) [58,59]. In contrast, the crystal structures of other P450s have revealed that such amino acid residues are not present on the distal side of their heme centers. Previous reports have suggested that the introduction of a basic residue can modify myoglobin into a peroxidase through site-directed mutagenesis [60,61,62]. Similar strategies have been applied to improve the peroxygenase/peroxidase activity of P450s [63,64]; however, the catalytic efficiency was not always satisfactory. Crystal structure studies have provided hints for the poor activity in some cases, namely, the basic group on the side chain of the introduced residue is distal from the heme center such that this residue cannot efficiently activate H_2_O_2_ as the general acid–base catalyst [63].

Based on previous reports, it has become clear that to use the shunt pathway of P450s efficiently, two points should be met: (1) a basic group located on the distal side of the heme center is necessary; and (2) the basic group should be placed at a suitable position to ensure that this residue plays the role as an acid-base catalyst efficiently. To this end, Ma et al. designed a DFSM approach to modify cytochrome P450BM3 monooxygenase from *Bacillus megaterium* into its peroxygenase mode (Figure 3) [65,66,67]. Typical structures of DFSMs, such as *N*-(ω-imidazolyl)-fatty acyl-l-amino acid (Im-Cn-AA), are shown in Figure 3B [66]. These DFSMs have an acyl amino acid moiety at one end as an anchoring group to bind with the enzyme, and an imidazolyl group at the other end as a basic group to assist the activation of H_2_O_2_. Moreover, the position of imidazolyl can be optionally tuned by changing the chain length of a flexible spacer having various CH_2_ numbers, while the introduced basic residues by site-directed mutagenesis often can’t extend sufficiently into a suitable catalytic site [63]. Ma et al. reasoned that the DFSM-facilitated P450BM3-H_2_O_2_ system was capable of running smoothly with a catalytic cycle that was similar to the native UPO peroxygenase under ideal conditions (Figure 3C) [68,69].

This concept was firstly validated by the H_2_O_2_-dependent epoxidation of styrene catalyzed by the P450BM3_F87A mutant. The presence of the best DFSM, *N*-(ω-imidazolyl)-hexanoyl-l-phenylalanine (Im-C6-Phe), increased the catalytic turnover number (TON) more than 30-fold than that of the F87A alone. The roles of the DFSMs were further demonstrated by using mono-functional small molecules (MFSMs) without the terminal imidazolyl group or acyl amino acid group, the latter didn’t improve TON and even inhibit the reactions. The ability of the DFSMs to generate peroxygenase activity was further demonstrated by using the double mutant F87A/T268V. The authors found that mutating the highly conserved T268 [69,70,71,72,73] abolished the H_2_O_2_ activity of the enzyme, which can be recovered upon the addition of DFSM. This discovery provides a unique choice of protein engineering sites for developing catalytic promiscuity of the current peroxygenase system (will be discussed below by combination with other results).

The catalytic role and mechanism of DFSMs have been further disclosed by combining structural biology and computational chemistry [74]. To mimic the pre-reaction state of P450-bounded H_2_O_2_ and avoid the H_2_O_2_-initiated reaction, Jiang et al. skillfully adopted the NH_2_OH molecule as the analog of H_2_O_2_ to prepare the co-crystal (Figure 4A,B). As a result, they successfully reported the first X-ray structure of the P450BM3 heme domain F87A mutant in complex with the DFSM, *N*-(ω-imidazolyl)-hexanoyl-l-phenylalanine (Im-C6-Phe) and NH_2_OH at 2.70 Å resolution (PDB ID: 7EGN, Figure 4C). The crystal structure clearly shows that Im-C6-Phe bound to P450BM3 through an H-bond network formed by interactions of its terminal carboxyl group with Arg47 and Tyr51, and hydrophobic interactions between its benzyl moiety and a hydrophobic pocket composed of Pro25, Val26, Leu29, Met185, and Leu188 (Figure 4D). The unique binding mode that involves additional hydrophobic interactions is distinct from those observed in the co-crystals of P450BM3 in complex with fatty acids (native substrates) or perfluoroacyl amino acids (decoy molecules) [75,76,77]. This binding mode plays a crucial role in positioning the imidazolyl group of the DFSM above the heme center, where the distance between the heme iron atom and the terminal nitrogen atom of the imidazolyl group is ~5 Å, indicating the imidazolyl group of the DFSM may act as a general acid–base catalyst in H_2_O_2_ activation, consistent with the original hypothesis by Ma et al. [66].

The mechanism for H_2_O_2_ activation was further elucidated by QM-MM computational investigations. These computational chemistry results revealed the crucial role of DFSM in promoting a heterolytic O–O cleavage to favor Cpd I formation [74]. The DFSM facilitates the formation of a proton channel between the imidazolyl group of the DFSM and proximal H of H_2_O_2_, thus enabling a heterolytic O–O cleavage and Cpd I formation, which is similar to the proposed mechanism for H_2_O_2_ activation in natural peroxygenases (e.g., UPO) or peroxidases (e.g., HRP). In contrast, the formation of Cpd I is apparently sluggish via the O–O homolysis mechanism in the absence of the DFSM. Similar results were also observed in the theoretical simulation of H_2_O_2_ activation by the P450cam T252A mutant [78], indicating weak H_2_O_2_ activation by NADPH-dependent P450s.

## 3. Catalytic Applications of the DFSM-Facilitated P450 Peroxygenase

In recent years, peroxygenase UPO has attracted considerable attention because of its versatile oxidation functions and potential in synthetic applications [79,80,81,82,83,84,85,86]. Moreover, peroxygenase that uses green and economic H_2_O_2_ to circumvent the use of expensive NADPH and the complex electron transfer system (redox partner proteins) has become a promising practical bio-oxidative catalyst when compared with using NAD(P)H-dependent P450 monooxygenases [79]. Despite concerns about the potential damage of H_2_O_2_ to enzymes, the use of a controlled fed-batch reactor or in situ-generating H_2_O_2_ has been demonstrated to enhance effectively the stability of peroxygenases through control of the H_2_O_2_ concentration in the reaction system, resulting in high catalytic turnovers [87,88,89,90,91]. Therefore, developing the catalytic potential of the artificial P450 peroxygenase is not only expected to expand the chemical space of P450 enzymes but also act as a beneficial supplement to the relatively scarce natural peroxygenase resources in nature. In fact, the DFSM-facilitated P450BM3-H_2_O_2_ system has shown versatile unique catalytic activity towards the peroxygenation reaction of various non-native substrates, such as epoxidation, hydroxylation, and sulfoxidation [66,92,93,94,95].

Asymmetric epoxidation of unfunctionalized olefins represents an important organic transformation to prepare optically pure epoxides [96,97,98,99]; however, the (*R*)-enantioselective epoxidation of styrene seems more difficult to achieve than the (*S*)-enantioselective reaction through either synthetic molecular catalysts or natural enzymatic bio-catalysts [100,101,102,103,104,105,106,107,108]. DFSM-facilitated P450BM3 peroxygenase enabled access to (*R*)-enantioselective epoxidation of unfunctionalized styrene and its derivatives (Figure 5). In view of the potential of the double mutant F87A/T268V in the (*R*)-enantioselective epoxidation of styrene in the presence of Im-C6-Phe, Zhao et al. systematically evaluated the effect of T268 residue and disclosed the roles of the T268 mutation in tuning activity and enantioselectivity of the NAD(P)H- and H_2_O_2_-dependent P450BM3 system, respectively [45]. Based on the more selective, but lower activity profile of the double mutant F87A/T268I (97% *ee*, TON = 335), a mutant library was constructed by introducing additional mutations at ten key residues around the substrate-binding pocket (Figure 5A). Two beneficial mutants were determined to give high (*R*)-enantioselective epoxidation of styrene (98% *ee*) with >4000 TONs (Figure 5B). This approach also gave modest to very good TONs (362–3480) and high (*R*)-enantioselectivities (95–99% *ee*) for the epoxidation of various styrene derivatives (Figure 5C), being comparable with the best (*R*)-enantioselective styrene monooxygenases, such as *Se*StyA from *Streptomyces exfoliatus*, *Aa*StyA from *Amycolatopsis albispora*, and *Pb*StyA from *Pseudonocardiaceae* reported recently [109,110]. The further semi-preparative scale experiments suggest its potential application in styrene epoxidation [92].

The direct hydroxylation of small alkanes to alcohols is a long-standing challenge because of the higher bond dissociation energies (BDE) of their C–H bonds when compared with that of the corresponding hydroxylated products, the latter easily leads to overoxidation [111,112]. Natural oxidizing enzymes, such as methane monooxygenase, soluble butane monooxygenase (sBMO), fungal peroxygenase (*Aae*UPO), and engineered P450s, are promising biocatalysts for the selective hydroxylation of small alkanes [76,77,93,113,114,115,116,117,118,119,120,121,122,123]. Recently, Chen et al. reported the peroxide-driven hydroxylation of small alkanes (C_3_–C_6_) by using engineered P450BM3 variants assisted by DFSMs [93]. Compared with some main results through enzymatic hydroxylation of small alkanes [116,117,118,119,120,121,122,123,124], DFSM-facilitated P450BM3 peroxygenase showed unique features and catalytic activities (Table 1). The hydrophobic mutation of T268 residue substantially improved the hydroxylation activities of small alkanes, which is distinct from NADPH-dependent P450 enzymes [94]. Here, the presence of the DFSM was critical for accomplishing the catalytic functions of engineered P450BM3 variants because the activity is completely lost in the absence of the DFSM. Two triple-mutants BM3_F87A/T268I/A184I and BM3_F87A/T268I/A82T showed the highest total turnover numbers (TTN) for the hydroxylation of propane and *n*-Butane (Entries 1–2 in Table 1), respectively, with better activity than *Aae*UPO, the only known H_2_O_2_-dependent native hydroxylase for small alkanes (entries 20–21 in Table 1) [116], and comparable activity to the P450BM3 decoy system (entries 3–8 in Table 1) [117,118,119,120], but far lower than P450_PMO_R1 and P450_PMO_R2, two evolved NADPH-dependent propane monooxygenases (entries 13–14 in Table 1) [121]. Notably, the product formation rates (PFR) for 2-propanol and 2-butanol of the current artificial P450 peroxygenase are far better than all reported natural or engineered enzyme systems. The contradiction between high PFR and low TTN suggests that the DFSM-facilitated P450 peroxygenase may be unstable. Nonetheless, reducing instability should yield an efficient biocatalyst for the direct hydroxylation of small alkanes. In addition, this peroxygenase system is unavailable for the hydroxylation of smaller alkanes (e.g., ethane and methane), which has been achieved by natural methane monooxygenase (MMO) or other enzymes (entries 9, 12, 16, 17 in Table 1) [76,77,118,122,123]. Anyhow, Ciuffetti et al. reported that CYP52L_1_ from *Graphium* sp. ATCC 58,400 can oxidize propane, but without any turnover numbers or catalytic constants mentioned [124]. This may be the only known P450 enzyme that uses gaseous alkanes as natural substrates, suggesting that P450 has a weak preference for small alkanes. Therefore, further protein engineering may be necessary for the DFSM-facilitated P450BM3 peroxygenase to access the direct hydroxylation of methane or ethane.

The *O*-demethylation of aromatic ethers is of important reaction to produce value-added phenolic compounds, which is also involved in aromatic ring-opening reactions of coniferyl and sinapyl lignin derivatives [125,126]. Various powerful oxidative enzymes, such as peroxidases from white-rot, soft-rot, and brown-rot fungi, as well as some bacteria, can catalyze demethylation of lignin-derived compounds and their model compounds [127,128,129,130]. A few P450 enzymes also show promise as an *O*-demethylase for lignin-derived aromatic ethers [131,132,133,134,135,136]. Recently, Jiang et al. successfully applied the DFSM-facilitated P450BM3 peroxygenase system to perform *O*-demethylation of various aromatic ether substrates (Table 2) [94]. These reactions show excellent regioselectivity toward the hydroxylation of the methoxy of aromatic ethers to give the demethylation product after automatically releasing formaldehyde. A suitable combination of the beneficial mutant and DFSM is important for controlling good regioselectivity. For example, some combinations examined still give aromatic hydroxylation as the main product. Although the DFSM-facilitated P450BM3 peroxygenase appears to open a new avenue for the key demethylation step in the bioconversion of lignin, it is still restricted by low TONs and narrow substrate scopes.

The DFSM-facilitated P450BM3 peroxygenase is also capable of catalyzing other types of reactions, including thioanisole sulfoxidation, aromatic hydroxylation of naphthalene, and the benzylic hydroxylation of ethylbenzene (Figure 6) [66,95]. The F87A mutant exclusively yielded sulfoxide with a PFR of 888 µmol·min^−1^·(µmol P450)^−1^ and a catalytic TON of 3436 in the presence of the DFSM, 35-fold higher than that without the DFSM. Interestingly, T268 mutations drastically affected the hydroxylation activity of the system, e.g., the double mutants F87G/T268V and F87A/T268V increased the TONs for the formation of 1-naphthol and 1-phenylethanol to 200 and 319, being around 15-fold and 8-fold to the single mutant F87G and F87A, respectively [95].

## 4. Switching Peroxidase Activity of the DFSM-Facilitated P450 Peroxygenase

The catalytic promiscuity of enzymes is a fascinating topic for the biochemistry, synthetic biology, and chemical biology communities [137,138,139]. P450s have been well documented to carry out multiple catalytic functions such as monooxygenase, peroxygenase, and peroxidase activity [140]. However, research interest has focused on the monooxygenase and peroxygenase activities of P450s, and only a handful of studies have examined the catalytic peroxidase functionality of P450s. The non-natural DFSM-facilitated P450-H_2_O_2_ system described above mainly catalyzes various per-oxygenation reactions, including epoxidation, hydroxylation, and sulfoxidation [66,92,93,94,95]. Interestingly, the oxidation of guaiacol, a classical substrate of peroxidases [141,142,143,144], catalyzed by the DFSM-facilitated P450BM3-H_2_O_2_ system yielded demethylated catechol as a major product, suggesting it mainly functioned as a peroxygenase but not as a peroxidase [94]. After carefully analyzing the catalytic mechanism of the potential competitive oxidation pathways in the DFSM-facilitated P450BM3-H_2_O_2_ system, Ma et al. hypothesized that mutation of redox-sensitive residues may enable switching of peroxygenase activity to peroxidase activity [145]. Using a semi-rational design approach, similar to FRISM (focused rational iterative site-specific mutagenesis) named by Reetz and Wu [146,147], Ma et al. identified mutations of three key redox-sensitive tyrosine residues that are located on the surface of P450. Screening for activity-enhanced peroxidase mutants yielded a mutant that efficiently catalyzed one-electron oxidation of guaiacol through combination with other redox-sensitive residues located in the electron transfer pathway. The engineered system also exhibits favorable one-electron oxidation activity toward other peroxidase substrates, including 2,6-dimethoxyphenol, *o*-phenylenediamine, and *p*-phenylenediamine, and almost without peroxygenase activity for these substrates. Notably, this system attains the best peroxidase activity of any P450 reported [56,148], and rivals most natural peroxidases [149,150,151,152,153], suggesting significant potential for catalytic promiscuity of the DFSM-facilitated P450BM3-H_2_O_2_ system (Figure 7). Future efforts should explore the functional applications of the DFSM-facilitated P450 peroxidase in synthetic chemistry.

## 5. Summary and Perspectives

In summary, although only a few natural P450s, such as CYP152 peroxygenases from *Sphingomonas paucimobilis*, *Bacillus subtilis*, and *Clostridium acetobutylicum* can directly use an oxygen atom from peroxides for oxidation reactions [42,43,44,45,46,47], the engineered artificial P450 peroxygenases have significantly expanded the substrate scope and reaction types of P450-catalyzed per-oxygenation reactions. Therefore, it is no exaggeration to state that H_2_O_2_-driven P450 peroxygenases are emerging as powerful bio-oxidation catalysts. Among these, the DFSM-facilitated P450 peroxygenases provide a novel and unique solution for the efficient use of H_2_O_2_ by P450s, which exhibit much higher H_2_O_2_ activities in various reactions when compared with those P450 peroxygenases that have been engineered through site-directed mutagenesis and directed evolution [26,27,28,56,63,64,77,152,153]. Moreover, the DFSM-facilitated P450 peroxygenases may offer better opportunities for enhancing the regio- and enantioselectivity in oxidation reactions of non-natural substrates. On the one hand, the introduced DFSMs can influence the orientation of substrates through interaction with each other to modulate reaction selectivity, besides its role in the activation of H_2_O_2_, which still requires further experimental characterization. On the other hand, the highly conserved T268 residue can be optionally mutated in the DFSM-facilitated P450 peroxygenase system. In contrast, the mutation of T268 is not favorable in NADPH-dependent P450BM3 oxidation because this residue is located on the proximal side of the heme center and is thought to play multiple roles in NADPH-dependent catalysis [70,71,72,73,154]. In fact, successful examples of the DFSM-facilitated P450 peroxygenase system have demonstrated that mutation of T268 has a significant influence on regulating the substrate pocket space when employing a protein engineering strategy [45,46,47,48]. This suggests that protein engineering of the DFSM-facilitated P450 peroxygenase system may also have its own unique advantages for controlling reaction selectivity in comparison with natural NADPH-dependent P450s. In addition, the high peroxidase activity of the DFSM-facilitated P450-system developed recently expanded the catalytic promiscuity of the system [145], whose further application in organic transformation is expected.

The unique selectivity and activity of the DFSM-facilitated P450 peroxygenase system have shown its potential to be as a promising bio-oxidative catalyst; however, it is worth noting that there are still some drawbacks to hamper its further industrially utilization: (1) despite high efficiency, the introduction of DFSM undoubtedly increases the cost of the catalytic reaction, especially when a large excess is required; (2) oxidative damage of P450 caused by the presence of a large amount of H_2_O_2_; (3) the uncertainty associated with applying this strategy to other P450s; (4) the complex structures of DFSMs lead to an increase in the threshold of popularization and use; (5) the full catalytic mechanism still needs to be elucidated. In conclusion, the DFSM-facilitated P450 peroxygenase system simultaneously faces opportunities and challenges. Maximizing the potential of the system and answering the above issues will open new avenues for developing P450-based biocatalysts.

## Figures and Tables

**Figure 1 antioxidants-11-00529-f001:**
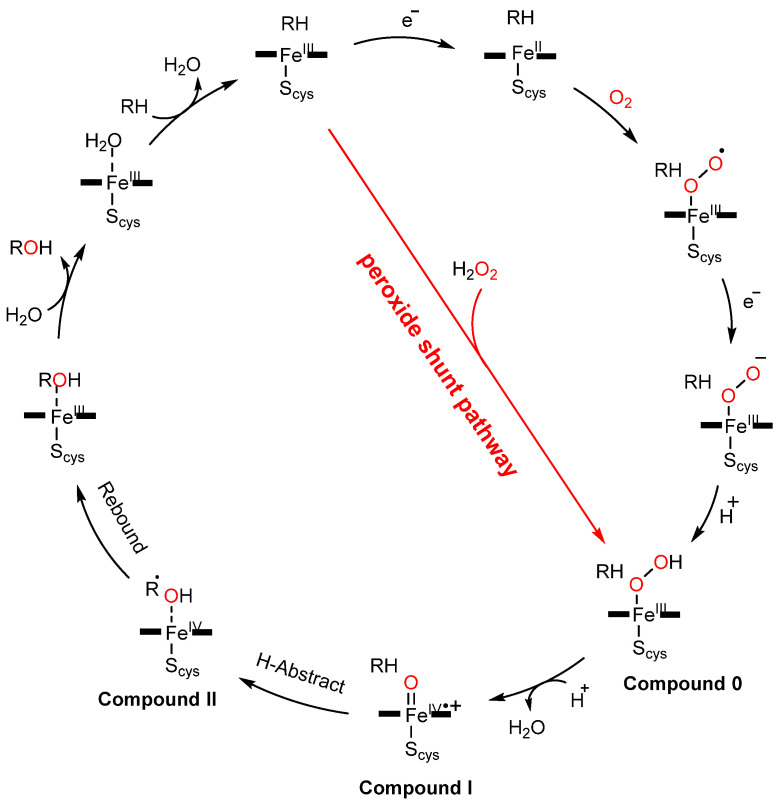
The catalytic cycle of cytochrome P450 monooxygenase and the peroxide-shunt pathway.

**Figure 2 antioxidants-11-00529-f002:**
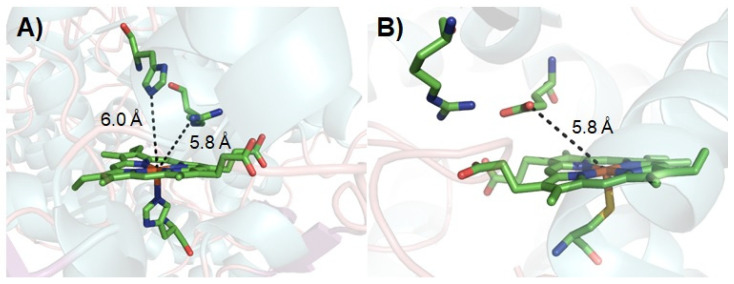
Active site structures of HRP ((**A**), PDB ID: 1ATJ) [58] and unspecific peroxygenase ((**B**), PDB ID: 2YOR) [59].

**Figure 3 antioxidants-11-00529-f003:**
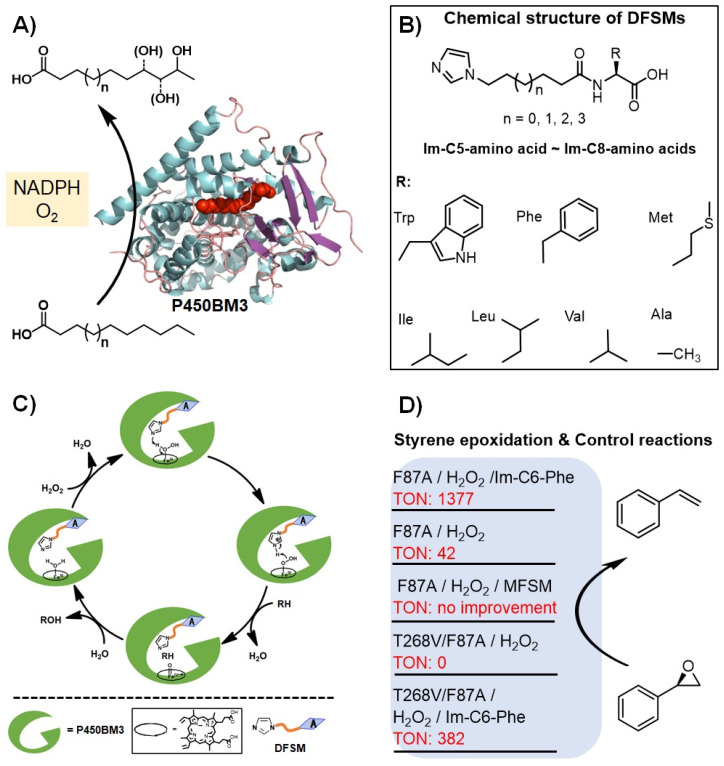
Proof-of-concept of the dual-functional small molecule (DFSM)-facilitated P450 peroxygenase. (**A**) The NADPH-dependent P450BM3 monooxygenase. (**B**) Proposed catalytic cycle of the DFSM-facilitated P450 peroxygenase. (**C**) Chemical structures of the DFSM molecules. (**D**) Styrene epoxidation in the presence of the DFSM and control experiments.

**Figure 4 antioxidants-11-00529-f004:**
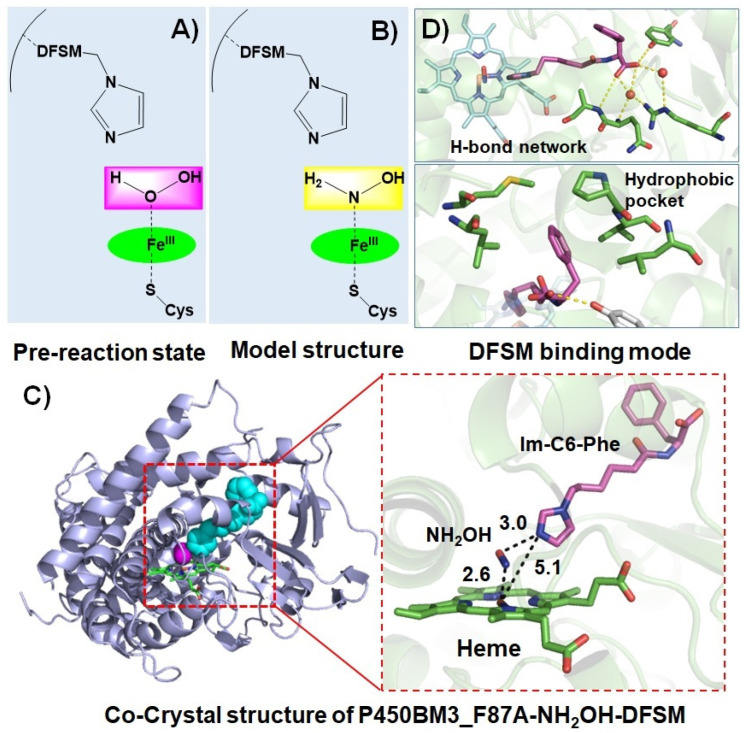
Structural basis of the DFSM-facilitated P450 peroxygenase. (**A**) Proposed pre-reaction state of P450BM3 in the presence of H_2_O_2_ and DFSM. (**B**) The model structure with NH_2_OH instead of H_2_O_2_. (**C**) The co-crystal structure of P450BM3_F87A in complex with NH_2_OH and Im-C6-Phe. (**D**) The binding interactions of Im-C6-Phe with P450BM3.

**Figure 5 antioxidants-11-00529-f005:**
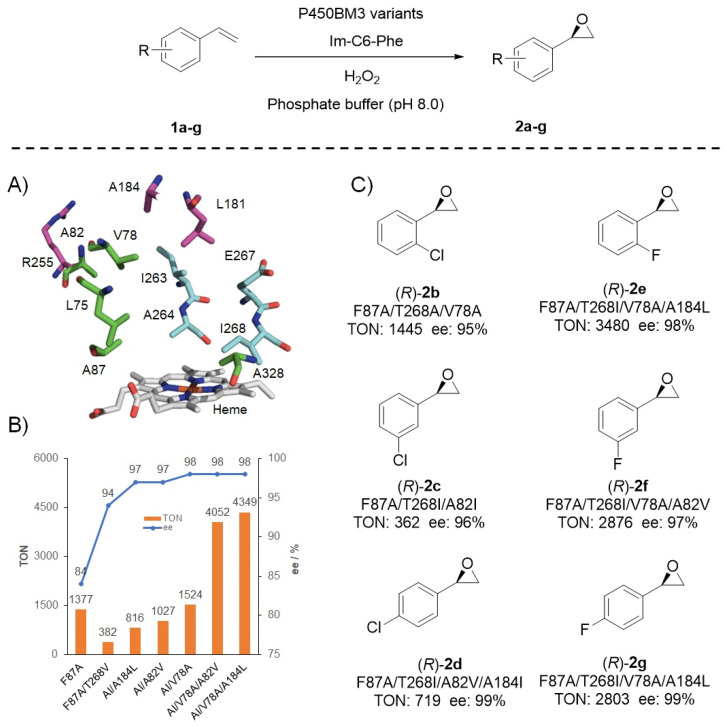
Protein engineering of the DFSM-facilitated P450BM3 peroxygenase for catalyzing (*R*)-enantioselective epoxidation of styrene and its derivatives. (**A**) Key residues around the substrate-binding pocket of P450BM3; (**B**) protein engineering for styrene epoxidation; (**C**) the epoxidation of styrene derivatives by the DFSM-facilitated P450 peroxygenases.

**Figure 6 antioxidants-11-00529-f006:**
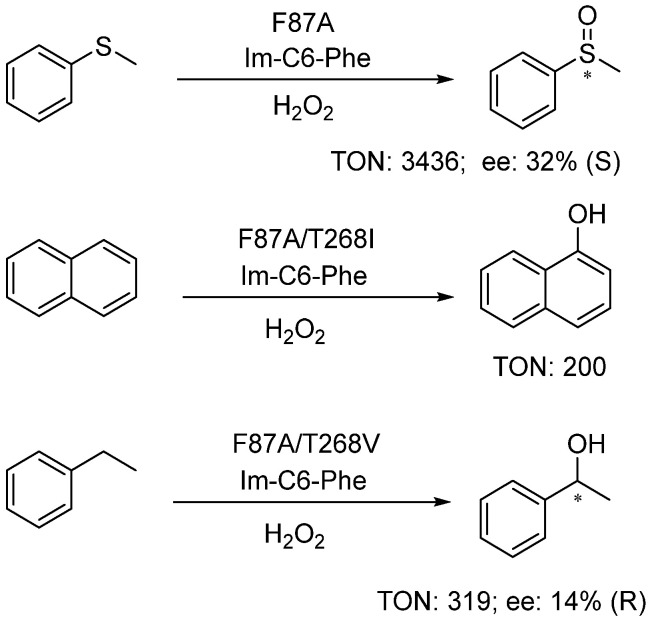
DFSM-facilitated P450 peroxygenases catalyzed sulfoxidation and hydroxylation.

**Figure 7 antioxidants-11-00529-f007:**
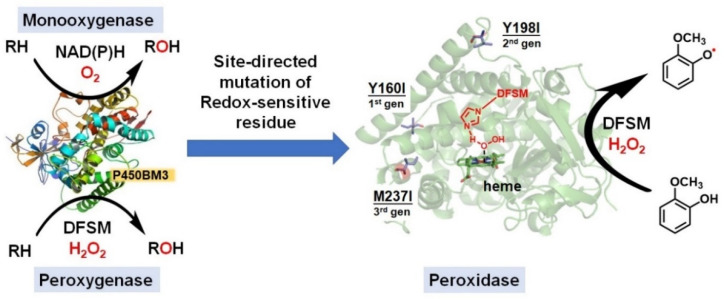
Native monooxygenase activity. DFSM-facilitated peroxygenase activity and switching to peroxidase activity by mechanism-guided protein engineering.

**Table 1 antioxidants-11-00529-t001:** Catalytic hydroxylation of small alkanes by various enzymes in literature.

Entry	Enzyme	Alkanes	Final Product	PFR ^a^	TTN ^b^	Ref.
1	BM3_F87A/T268I/A184I/Im-C6-Phe	Propane	2-Propanol	630	1775	[93]
2	BM3_F87A/T268I/A82T/Im-C6-Phe	*n*-Butane	2-Butanol	1042	2253	[93]
3	BM3/PFC10	Propane	2-Propanol	70	700	[117]
4	BM3/PFC9-L-Leu	Propane	2-Propanol	256	2560	[76]
5	BM3/3CCPA-Pip-Phe	Propane	2-Propanol	615	-	[118]
6	BM3/PFC9	*n*-Butane	2-Butanol	110	1100	[120]
7	BM3/PFC11	Propane	2-Propanol	-	1021	[120]
8	BM3/PFC7	*n*-Butane	2-Butanol	-	3632	[120]
9	BM3/C7AM-Pip-Phe	Ethane	Ethanol	82.7	-	[118]
10	P450cam_EB	*n*-Butane	2-Butanol	520	-	[123]
11	P450cam_EB_L294M/T185M/L1358P/G248A	Propane	2-Propanol	505	-	[123]
12	P450cam_EB_L294M/T185M/L1358P/G248A	Ethane	Ethanol	78.2	-	[123]
13	P450_PMO_R1	Propane	2-Propanol	455	35,600	[121]
14	P450_PMO_R2	Propane	2-Propanol	370	45,800	[121]
15 ^c^	CYP52L_1_	Propane	1-Propanol	-	-	[124]
16	sMMO	Methane	Methanol	78	-	[122]
17	sMMO	Ethane	Ethanol	45.6	-	[122]
18	sMMO	Propane	2-Propanol	33–58.8	-	[122]
19	sMMO	*n*-Butane	2-Butanol	7.2–28.8	-	[122]
20	*Aae*UPO	Propane	2-Propanol	17	999	[116]
21	*Aae*UPO	*n*-Butane	2-Butanol	21	1258	[116]

^a^ PFR: product formation rate in µmol·min^−1^·(µmol P450)^−1^. ^b^ TTN: total turnover number. ^c^ There is no catalytic turnover data reported.

**Table 2 antioxidants-11-00529-t002:** Regioselective aromatic O-dealkylation by the DFSM-facilitated P450 peroxygenases.

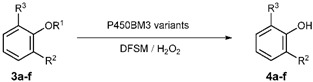								
Substrate				Enzyme System	Product			TON
No.	R^1^	R^2^	R^3^		No.	R^2^	R^3^	
**3a**	Me	H	H	BM3_F87A/T268I/Im-C5-Phe	**4a**	H	H	486
**3b**	Me	Me	H	BM3_F87A/T268I/Im-C6-Phe	**4b**	Me	H	356
**3c**	Me	OH	H	BM3_F87A/Im-C6-Phe	**4c**	OH	H	539
**3d**	Et	OH	H	BM3_F87G/T268G/Im-C5-Phe	**4d**	OH	H	99
**3e**	Me	OMe	H	BM3_F87A/T268I/Im-C6-Phe	**4e**	OMe	H	287
**3f**	H	OMe	OMe	BM3_F87G/T268V/Im-C5-Phe	**4f**	OH	OMe	165

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
