# Peer review of "A Unique P450 Peroxygenase System Facilitated by a Dual-Functional Small Molecule: Concept, Application, and Perspective"

_antioxidants, 2022, doi:10.3390/antiox11030529_

Round 1

Reviewer 1 Report

This review describes the use of CYP-peroxygenase systems as new bio-catalysts that might be useful in chemical synthesis.

The paper is well written with nice diagrams; however, it repeats many statements and descriptions already published by the last author, in particular the reference 30 by Ma et al. (2018). This review should be focused on the novelties and applications and could be more concise.

Author Response

Thanks for the valuable comments. We have made efforts to re-write some paragraphs to reduce the length of the paper, in particular, those related to the reference 30. In addition, we also added description to emphasize the novelties and applications of this new system.

Reviewer 2 Report

This is very interesting review summarizing the process of developing artificial P450

peroxygenase system.

I have the following remarks:

1) in the middle of the text there is no need to use capital letters;

2) Line 352: “Despite the many advantages of the DFSM-facilitated P450 peroxygenase system

worth exploring,……..” – unclear, what was the intention of Authors?

3) reference style should be adjusted to the requirements of the Journal.

Author Response

Thanks for your valuable comments! Below is our point-to-point responses:

  • We revised the misused capital letters.
  • We re-wrote this sentence as “The unique selectivity and activity of the DFSM-facilitated P450 peroxygenase system has shown its potentials to be as a promising bio-oxidative catalyst, however, it is worthy to note that there are still some drawbacks to hamper its further industrially utilization”.
  • We revised reference style.

Reviewer 3 Report

I think this is a nice review on the small molecule-assisted activation of P450 peroxygenase. The topic is well-organized and written with enough details. I enjoy reading this review and learned a lot myself!

Author Response

Many thanks for positive comments!